# Peritoneal Dialysis and the Role of Exercise Training Interventions

Osasuyi Iyasere [1,2,*] , Hannah M. L. Young [3,4] and James O. Burton [1,2]

1   John Walls Renal Unit, Leicester General Hospital, University Hospitals of Leicester NHS Trust, Leicester LE5 4PW, UK; jb343@le.ac.uk
2   Department of Cardiovascular Sciences, University of Leicester, Leicester LE1 7RH, UK
3   Lifestyle and Health Group, Leicester Diabetes Centre, University Hospitals of Leicester NHS Trust, Leicester LE5 4PW, UK; hy162@leicester.ac.uk
4   Department of Respiratory Sciences, University of Leicester, Leicester LE1 7RH, UK
*   Correspondence: osasuyi.iyasere@nhs.net

**Abstract:** People receiving peritoneal dialysis (PrPD) tend to be physically inactive, with consequent adverse outcomes including increased mortality, reduced technique, and hospitalization free survival. Exercise is a form of planned physical activity which has the potential to improve these outcomes. Feasibility studies suggest that exercise interventions are safe in PrPD. However, the uptake of exercise is low. In this review, we explore the benefits of exercise in this population, noting the limitations in the existing evidence. We highlight the challenges and uncertainties associated with exercise, including the perceptions of patients and clinicians. Finally, the opportunities for increasing exercise uptake are discussed, alongside future research priorities.

**Keywords:** exercise; physical activity; peritoneal dialysis

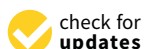



## 1. Introduction

In the UK, 5.4% of people receiving renal replacement therapy (RRT) receive peritoneal dialysis (PD) [1]. PD has the advantage of being a home-based therapy and is often chosen for the flexibility and independence it offers [2]. Nevertheless, people receiving PD (PrPD), as with other RRT modalities, are burdened by adverse outcomes including cardiovascular disease, mortality, multimorbidity, and lower levels of physical function and quality of life [3,4].

Similarly, levels of physical activity, defined as "any bodily movement produced by skeletal muscle which results in caloric expenditure" [5], are reportedly low in the PD population, with only 6 to 9% of PrPD reporting recommended levels of physical activity, according to observational studies [3,6,7].

Lifestyle interventions, particularly exercise, (a subset of physical activity that is planned, structured, and repetitive), offer potential benefits to PrPD, given that low physical activity is linked to poor clinical outcomes. Observational research suggests that reduced exercise capacity is associated with significantly reduced technique survival, peritonitis, and hospitalization free survival, in addition to increased mortality [8–11].

In this review, we explore the feasibility, benefits, uncertainties, opportunities, and challenges associated with exercise in PrPD and suggest future research priorities.

## 2. Evidence for the Feasibility and Benefits of Exercise for People Receiving PD

### 2.1. How Feasible and Acceptable Is Exercise in People Receiving PD?

The current evidence indicates that exercise interventions can feasibly be delivered in PrPD. Watanabe et al. [12] evaluated the impact of six months of home-based exercise comprising of walking, resistance exercises, and stretching on bone mineral density in PrPD. In this randomized pilot study of 53 Japanese PD patients, exercise was supervised,

and adherence encouraged using exercise diaries, monthly hospital visits and fortnightly contact with a study completion rate of 82%.

Bennett et al., evaluated a similar exercise program in a three-month randomized feasibility study of 36 PD patients, finding that exercise was feasible and safe. The program was also supervised with regular telephone contact and monthly visits, resulting in a completion rate of 72% [13]. In another randomized trial of 47 PD patients, 92% of the intervention group completed the 12-week exercise program. This comprised of aerobic and resistance exercises supervised using weekly postcards to assess adherence and monthly visits. The adherence rate for the aerobic and resistance exercises was 52% $\pm$ 40% and 76% $\pm$ 37%, respectively. No adverse events were reported [14].

These studies suggest that exercise interventions are feasible and safe in PrPD. A common driver for adherence appeared to be regular exercise supervision by suitably trained staff. The lack of exercise professionals in the renal multidisciplinary team has been identified as a barrier to the development of exercise programmes for patients with CKD [15]. Furthermore, the follow up period for these studies was short, with limited information on adherence to exercise beyond the intervention period. It is therefore unclear how sustainable such interventions are and whether they can be delivered in real world healthcare systems.

Furthermore, while exercise has been shown to be feasible in frail HD patients [16], data on the feasibility of exercise training in frail PD patients is lacking. Nevertheless, recent guidance suggests that walking or stationary cycling, body weighted resistance exercises, and balance exercises be considered in this group [17].

### 2.2. What Are the Benefits of Exercise in People Receiving PD?

2.2.1. Cardiovascular Benefits

Cardiovascular disease remains the leading cause of death in PrPD [18]. Exercise has been shown to improve some cardiovascular parameters in patients with CKD [19] and on HD [20], including blood pressure, RR interval variability, and left ventricular mass. However, there are limited data on the impact of exercise on cardiovascular markers in PrPD. A 12-week program of structured aerobic exercise did not significantly alter left ventricular diastolic and systolic diameters or ejection fraction in a study of 13 people receiving CAPD [21].

2.2.2. Physical Function

Exercise may improve measures of physical function in PrPD. A six-month home-based exercise was evaluated in a randomized study of 53 patients on PD (median age—intervention 66 years; control 64 years), the thirty second chair-stand, 6 min walk tests improved significantly in the intervention group [12]. These findings have been corroborated in a larger study of people receiving HD and PD [22]. Exercise capacity has also been shown to improve with exercise training in dialysis patients [23]. This is relevant, as exercise capacity has been associated with QoL [24], technique survival, peritonitis free survival, and PD related hospitalization free survival [8,25].

2.2.3. Bone Mineral Density

Exercise training consisting of high-impact multi-directional weight-bearing or resistance exercise is associated with higher bone mineral density (BMD), although the precise prescription for optimal benefit is less clear in older adults [26]. Whilst there is preliminary evidence to suggest similar programs are of benefit to the CKD population [27], this is less certain in PrPD. In the randomized study that evaluated the impact of a six-month home-based exercise program in 53 PrPD, there was no improvement in BMD [12]. The authors suggest that the short duration of the exercise intervention may explain the findings. In another observational study of 34 PrPD, moderate to vigorous physical activity correlated negatively with BMD [28]. The small sample size and cross-sectional nature of the study suggests that these results should be cautiously interpreted.

### 2.2.4. Residual Renal Function

Residual renal function is a key determinant of clinical outcomes in PrPD [29]. Exercise may slow GFR decline in patients with CKD [30]. Exercise may similarly be reno-protective in PrPD. In a post hoc analysis of an RCT evaluating 12-week home-based program of resistance and aerobic exercises, the intervention group has significantly lower urinary biomarkers of kidney disease progression. The difference in weekly residual creatinine clearance did not reach statistical significance, as the study was under powered [31].

### 2.2.5. Metabolic Effects

PD is associated with excessive glycemic load, which may result in adverse metabolic effects [32]. There is an uptake of between 100 and 300 g of glucose in 24 h of CAPD therapy. The consequences include hyperglycaemia, increased insulin requirement and visceral fat, weight gain, and metabolic syndrome [32]. It has been speculated that exercise may help counteract the metabolic effects associated with PD. Small studies have shown that aerobic exercise is associated with improved fasting glucose levels and a trend towards higher HDL cholesterol levels in PrPD [33,34]. In tandem with dietary and dialytic strategies to reduce glycaemic load, exercise has been associated with weight loss in a feasibility study of eight overweight PD patients [34]. High intensity interval training may also reduce in inflammation, a risk factor for cardiovascular mortality, in PrPD [35].

### 2.2.6. Quality of Life and Mental Health

Exercise has been shown to improve QoL in PrPD. In the randomized trial undertaken by Uchiyama et al. [14], the physical role and emotional functioning subscales of the KD QoL-SF significantly improved after a three-month exercise program. A pooled analysis of two interventional studies (*n* = 30) showed that exercise was associated with an improvement in SF 36 scores across all domains [36]. A pilot study found that the mental health domain of SF-36 improved after a three-week program in six patients on PD [37]. Exercise may also have a positive effect on depression scores [38]. In a randomized trial of 70 PrPD, a 12-week Baduanjin exercise program was associated with improved QoL, assessed using the KD QoL-SF. This exercise intervention, which is of low aerobic intensity, was associated with an improvement in the effect of kidney disease subscale in the intervention group. The physical and mental component scores were also significantly higher at 12 weeks when compared to the control group [39].

In a multicenter study of 247 HD and 49 PD, the cognitive subscale of the KD QoL-SF improved in those who completed a six-month walking exercise program, in contrast to the control group [22].

### 2.2.7. Symptoms Associated with Peritoneal Dialysis

Fatigue is a common symptom in PrPD and is often a perceived barrier to exercise [40,41]. A pilot study evaluating the effects of exercise on perceived fatigue in PrPD did not show a significant improvement following an eight-week program [42].

Exercise may improve the lumbosacral strain associated with an intraperitoneal dwell. In an exercise program of pelvic tilt, abdominal curls, and leg exercises in 11 PrPD, lumbosacral strain was ameliorated by improving muscle strength in the lumbosacral region [43].

## 3. Challenges and Uncertainties

There are some specific challenges and uncertainties that PrPD face when considering exercise, which are summarized within Figure 1.

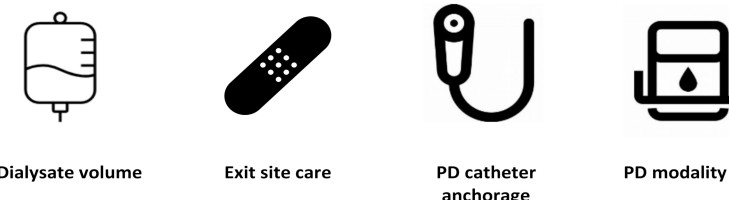

**Figure 1.** Factors to consider with exercise in PrPD.

### 3.1. Dialysate Volume

Intraabdominal pressure (IAP) positively correlates with dialysate volume and is also affected by posture in PD patients [44]. IAP is comparatively higher in sitting and standing positions than in the supine [45]. High IAP (typically greater than 16 to 20 cm $H_2O$) has been associated with a higher risk of mechanical complications including leaks and hernia [45]. Resistance exercise such as weightlifting also increases IAP in the general population [46]. There is also limited evidence that IAP increases in PD patients during physical activities such as straining, coughing and resistance training, potentiated with higher dialysate volumes [47]. One may therefore speculate that resistance exercise may increase the risk of abdominal discomfort, hernia formation, or leaks by increasing IAP. Therefore, the patient on continuous ambulatory PD or automated PD with a day dwell is typically advised to avoid strenuous resistance exercise or to do so with a "dry" abdomen. PD leaks are more common in the first few weeks after catheter insertion. Therefore, avoiding resistance exercise during the "break in" period may be considered for the same reasons. There is however a lack of robust evidence demonstrating a significant and sustained rise in IAP with exercise in PD patients that leads to complications. A feasibility trial of 36 PD patients implemented a three-month exercise program that included resistance exercise. Only one patient reported mild abdominal discomfort during a core exercise [13].

Raised intra-abdominal pressure has also been identified as a risk factor for pelvic organ prolapse in women receiving PD [48–50]. Whilst prolapse is rare, pelvic floor dysfunction, an antecedent to this condition, may be more common, but under-reported, in this population [49]. The early initiation of pelvic floor exercises may prevent pelvic floor dysfunction in those new to PD and improve outcomes for those with existing pelvic floor dysfunction [51]. This is particularly important given that management options for women that develop subsequent prolapse are limited [48,50]. However, there is a lack of evidence exploring pelvic floor dysfunction in women receiving PD [52].

### 3.2. PD Catheter and Exit Site Care

Avoiding trauma and traction to the PD catheter are key components of exit care [53]. Certain forms of exercise, such as contact sports, may trigger such events. The consequences may be extrusion of the external cuff on the PD catheter, a breach in the skin integrity around the exit site that potentiates the risk of infection or on rare occasions, catheter rupture [54]. It is also recommended to avoid the exit site being submerged in water at least in acute post catheter insertion phase and until the exit site heals. This has implications for those who wish to swim, due to the potential for exposure to infective pathogens.

Despite these concerns, worries about trauma or traction can be managed by education to ensure the PD catheter is secured appropriately prior to exercise and using protective gear where necessary. There is insufficient evidence to preclude PD patients from swimming [55]. Expert consensus recommends swimming only in seawater or private chlorinated pools using a waterproof dressing or colostomy bag over the exit site, as well as performing exit site care after swimming [17]. It does also seem sensible to avoid swimming early post PD catheter insertion or in the setting of infection or a skin breach at the exit site until it is completely healed. The evidence supporting these practice points are however limited.

*3.3. PD Modality*

Automated PD (APD) is typically undertaken overnight, leaving daytime hours free for other activities, in contrast to continuous ambulatory PD (CAPD). Consequently, whilst levels of physical activity do not appear to differ between CAPD and APD [10,11], it is plausible that the uptake of exercise may differ between CAPD and APD. To date, there are no studies comparing exercise uptake or outcomes between modalities.

**4. Are There Guidelines on Exercise for PrPD?**

The evidence supporting exercise within this population is clearly limited by small sample sizes and highly selected populations. A recent systematic review assessing the impact of exercise training in the dialysis population, could not draw robust conclusions in PrPD [56]. Indeed, recently published exercise and lifestyle guidance from the UK Kidney Association has does not include recommendations for PrPD [57]. The International Society of Peritoneal Dialysis (ISPD) and the Global Renal Exercise Network (GREX) has now produced practice points for physical activity in PrPD [17]. The key practice points range from the timing of physical activity to appropriate dietary strategies (Figure 2). While this is welcome as a platform for promoting exercise in this population, most of the practice statements are based on consensus expert opinion and low-grade evidence. Moreover, they focus predominantly on how to exercise safely and less so on evidence-based exercise regimes, emphasizing the need for high quality research in this domain.

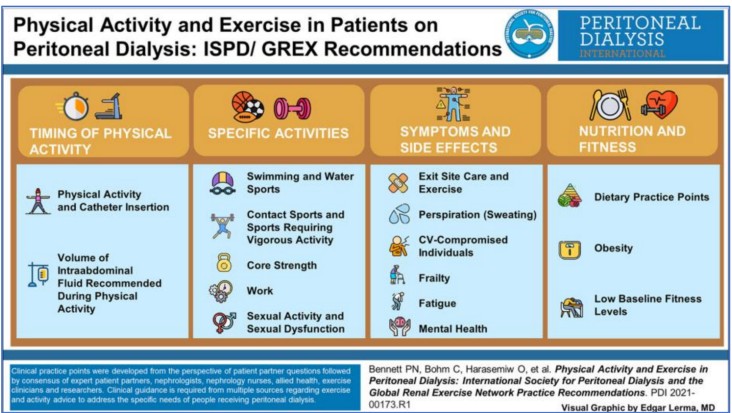

**Figure 2.** Exercise practice points for PrPD from the ISPD/GREX guidance (reprinted with permission from ref. [17]. 2021 Bennett, P.N., Bohm, C., Haraseniw, et al.).

**5. Patient and Clinician Perspectives**

There are perceived barriers to exercise in PrPD [58], despite the potential benefits. Patients, caregivers and health care professionals are key stakeholders in improving exercise uptake with PD and understanding their perspectives may well inform strategies to enhance exercise adherence.

*5.1. Patient Perceptions*

Three cross-sectional studies have evaluated patient perspectives on exercise in PrPD [40,41,59]. Two of these studies evaluated patient perceptions using the Dialysis Patient-Perceived Exercise Benefits and Barriers Survey. Exercise is generally perceived to be beneficial by PrPD in these studies. The commonly reported benefits are highlighted in Table 1. However, there are also several perceived barriers to the uptake of exercise (Table 2). The results from these three studies suggest that concerns about peritoneal dialysate and the PD catheter damage were not commonly perceived barriers. In addition, the baseline level of physical activity did not influence perceived barriers to exercise. Symptoms including tiredness, body pain, and muscle fatigue were more commonly reported barriers to exercise (alongside lack of information, concerns about safety, and low motivation).

**Table 1.** Common patient perceived benefits of exercise [40,41,59].

| Perceived Benefits of Exercise by PrPD | |
|---|---|
| Confidence | 11.1% |
| Improves sleep | 14.8% |
| Vitality | 16.7–66% |
| Independence | 69.2% |
| Delays decline in body function | 71.8% |
| Improves Mood | 72% |
| Weight control | 72–76.9% |
| Better quality of life | 79% |
| Prevents muscle wasting | 84.6% |

**Table 2.** Patient perceived barriers and enablers to exercise [40,41,59].

| Barriers | Enablers |
|---|---|
| Fatigue (64–69.2%) | Clinician encouragement (19.6–19.7%) |
| Muscle fatigue (43.6%) | Support from family and friends (18.5%) |
| Body pain (15.6–43.6%) | Participating in an exercise program (15.5%) |
| Worry about falling (33.3%) | |
| Lack of exercise information (25.6%) | |
| Poor motivation (24.4%) | |
| Fluid in the abdomen (23.3%) | |
| PD catheter concerns (10%) | |
| Low mood (5.2%) | |
| Fear of injury (4.3%) | |

The findings of these studies need to be interpreted with caution. Two of these studies evaluated HD and PD patients concurrently using tools originally developed and validated in the HD population [60] which do not necessarily capture PD specific themes. Zeng et al. [34] adapted this tool by adding two PD specific questions [40] and evaluated a small cohort of PD patients (*n* = 39) with findings that may not be generalizable. There may well be barriers specific to this population that are yet to be identified.

*5.2. Clinician and Caregiver Perspectives*

Support from doctors and nurses has been highlighted as an enabler for exercise in PrPD (Table 2) [40]. In a survey of Canadian dialysis patients, approximately 60% of PrPD indicated that they would exercise if advised to do so [61]. However, the prevalence of exercise counselling among nephrologists is reportedly low. In a cross-sectional survey of 550 nephrologists, only 38% reported counselling dialysis patients to increase physical activity [62]. A follow up survey undertaken six years later reported no change in counselling practices [63]. The perceived barriers to counselling include lack of time and confidence to discuss exercise, lack of belief that patients will respond, and a belief that other health conditions take precedence over exercise [38]. The views of the wider multidisciplinary team are relevant in designing a bespoke exercise intervention for PrPD. Encouragement from dialysis nurses, who may well have more patient contact compared nephrologists, has been identified as an enabler for exercise in this population [40]. There is a paucity of research evidence evaluating the perspectives of the wider multidisciplinary team.

There is even less research into caregiver perspective and exercise in PrPD. PD has been associated with caregiver burden. A cross sectional study by Zhang et al. [64] evaluated caregiver burden in a cohort of 170 PrPD and caregivers, respectively, with 78.2% reported at least mild caregiver burden. A higher level of frailty in PrPD was associated increasing caregiver burden. As frail patients tend to be less active, one may speculate that lower physical activity may be linked to caregiver burden. Interestingly, regular exercise by caregivers was associated with lower caregiver burden [64]. Intradialytic exercise has been linked to lower caregiver burden in HD patients [65]. The impact of exercise training in PrPD on caregiver burden has not been evaluated. Furthermore, there is evidence of benefit

with a dyadic approach to exercise in older adult caregivers and care recipients [66]. It is not certain how effective this strategy would be for PrPD.

## 6. Opportunities to Improve Exercise Uptake in PrPD

Increasing age, hemoglobin level, and volume status are known associates of low exercise capacity in PrPD and impact upon physical performance and engagement with exercise [67]. There is therefore additional incentive to treat anemia and fluid overload as a part of routine clinical care.

Efforts to combine exercise and physical activity programs with behavior change techniques, symptom management strategies, and education programs may be important to counteract the commonly perceived barriers to increased physical activity and exercise. Such programs would require appropriate workforce planning which factors in the role of exercise professionals in the multidisciplinary team [68]. They would ideally be tailored to patient preference. For example, PrPD may predominantly prefer to exercise at home with a program that includes aerobic and resistance exercise [61].

The COVID pandemic has heralded an expansion in the use of online platforms for healthcare delivery. Digital healthcare intervention platforms may therefore be instrumental in improving physical activity in PrPD. The Kidney Beam platform has recently been developed and launched as an online resource to promote physical activity in people living with CKD, including PrPD. It is hoped that the associated Kidney Beam trial will provide insights on its clinical and health economic value [69].

## 7. Future Research Priorities

Despite the potential of exercise to improve outcomes for PrPD, much uncertainty persists regarding exercise in this population. High quality research studies are needed to ascertain the relationship between exercise and catheter related infections. The relationship between core exercises, IAP, dialysate volume, and the risk of hernia formation or leaks warrants further study. Concerns about the potential risks with core exercises forms the basis for current guidance on leaving the peritoneal cavity dry during such activities. Conversely, core exercises may have a role in strengthening the abdominal wall to reduce the risk of hernias. Little is also known about the role of pelvic floor exercises in mitigating the risk of pelvic organ prolapse in women receiving PD.

The long-term benefits of exercise in PrPD including cardiovascular benefits, impact on frailty status, falls risk, transplant waitlist status, and life participation are largely unknown, as is the optimal exercise program for frail or multimorbid PrPD. Furthermore, there is limited evidence on suitable exercise recovery strategies in this population. As CKD is associated with accelerated aging [70], one may speculate that muscle recovery would be slower after exercise, as has been reported in older adults [71]. The ISPD/GREX recommendations on physical activity include advice on tailored fluid intake during and after exercise. There is also a suggestion to consume 20 g of high-quality protein after resistance exercise. However, these interventions have mostly been extrapolated from the general population [17].

While we have some insight into the barriers and enablers to exercise in PrPD, there is a dearth of studies using qualitative methodology that explore these identified themes. Furthermore, caregiver perspectives on the benefits, barriers, and enablers to exercise are yet to be explored.

The development of a suitable and sustainable exercise program for PrPD will require a multidisciplinary approach and would therefore require substantial resources. Such an exercise program may involve the use of digital health platforms. A health economic analysis is needed to evaluate the costs and potential financial benefits associated with its implementation.

## 8. Conclusions

Exercise training is a feasible in PrPD, but the benefits of such programs remain unclear, and the uptake of exercise remains low. Recent advances in the field, particularly published guidance, provide a foundation for standardizing and promoting exercise with PrPD. However, there remains a need for high quality research which focuses on establishing the safety and efficacy of exercise, elucidating how exercise should be tailored to this group and a better understanding of the perspectives of PrPD and the wider MDT.

**Author Contributions:** Writing—original draft preparation, O.I.; writing—review and editing, O.I., H.M.L.Y., and J.O.B. All authors have read and agreed to the published version of the manuscript.

**Funding:** H.M.L.Y. is supported by grants from the National Institute for Health Research (Funding number—NIHR301593). The views expressed in this publication are those of the authors and not necessarily those of the NHS, the NIHR or the Department of Health and Social Care.

**Institutional Review Board Statement:** Not applicable.

**Informed Consent Statement:** Not applicable.

**Data Availability Statement:** Not applicable.

**Conflicts of Interest:** The authors declare no conflict of interest.

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
