# Peer review of "Peritoneal Dialysis and the Role of Exercise Training Interventions"

_kidneydial, doi:10.3390/kidneydial2010007_

Round 1

Reviewer 1 Report

In this manuscript, the authors reviewed a number of published reports on the usefulness of exercise in people undergoing peritoneal dialysis (PrPD), highlighted the challenges and uncertainties associated with exercise, and concluded that although exercise training is feasible in PrPD, the benefits of such programmes remain unclear, and the uptake of exercise remains low.

This review is well-researched and well-written with citations to the recent papers. I don't think any major revisions are necessary, but it would be better to cite the following papers.

  1. Raimundo A, et al. High Levels of Physical Activity May Promote a Reduction in Bone Mineral Density in Peritoneal Dialysis. Medicina (Kaunas). 2020 Sep 11;56(9):464.
  2. Zhang F, et al. Effects of Baduanjin Exercise on Physical Function and Health-Related Quality of Life in Peritoneal Dialysis Patients: A Randomized Trial. Front Med (Lausanne). 2021 Nov 29;8:789521.
  3. Bernier-Jean A, et al. Exercise training for adults undergoing maintenance dialysis. Cochrane Database Syst Rev. 2022 Jan 12;1(1):CD014653.

Author Response

Thank you for your suggestions. These references have been included in the manuscript.

Reviewer 2 Report

Iyasere et al present a review article entiled "Peritoneal dialysis and the role of exercise training interventions". The authors discuss the opportunities for increasing exercise uptake in CKD patients receiving peritoneal dialysis (PrPD) that are mostly physically inactive, with consequent adverse outcomes.

The review is well written, original, and of importance in the field.

IN the chapter "benefits of exercise in people receiving PD"

I woudl advise to also include and discuss anemia burdening in the patients. You can see PMID: 32899941 for a review on biology and symptoms. Known causes of muscle wasting and reduced physical functioning in CKD include uraemic myopathy and neuropathy, inactivity, and anaemia.

Figures 1 and 2 are a bit poor. More information (such as percentages..) should be given.

In discussion I would advise to discuss recovery protocols that can be used after the physical exercices in these fragile patients. Not a lot is known in CKD patients but the authors can refer to studies that were performed on elderly patients.

Minor

correct "PD has the advantage OF being a home-based therapy"

What does "reduced exercise capacity is associated with significantly reduced technique" mean?

rewrite "Another randomised trial of 47 PD patients, 92% of intervention
group completed the 12-week exercise program. This comprised of aerobic and resistance exercises"

Author Response

Thank to you for the review and suggestions. We have made some revisions as follows

  • We the suggested paper on anaemia with interest. However it seemed to focus largely on the pathophysiology and management of anaemia.  We were ot certain how  this would fit into the exercise benefit section. We have however commented on the need to manage anaemia under the opportunities section, being that haemoglobin level is a clinical associate for  exercise capacity.
  • We have changed figure 1 and converted the original figure 2 (perceived benefits)  to a table with percentages
  • We have included a few sentences on recovery strategies in the future priorities section. It is noted that any existing advice has been extrapolated from the general population.
  • We have addressed the typographical errors highlighted